# A 20-Questions-Based Binary Spelling Interface for Communication Systems

**DOI:** 10.3390/brainsci8070126

**Published:** 2018-07-02

**Authors:** Alessandro Tonin, Niels Birbaumer, Ujwal Chaudhary

**Affiliations:** 1Institute of Medical Psychology and Behavioral Neurobiology, University of Tübingen, 72076 Tübingen, Germany; alessandro.tonin@uni-tuebingen.de (A.T.); niels.birbaumer@uni-tuebingen.de (N.B.); 2Wyss-Center for Bio- and Neuro-Engineering, 1202 Geneva, Switzerland

**Keywords:** brain computer interface, complete locked-in state, communication, Artificial Neural Network, 20-questions-game

## Abstract

Brain computer interfaces (BCIs) enables people with motor impairments to communicate using their brain signals by selecting letters and words from a screen. However, these spellers do not work for people in a complete locked-in state (CLIS). For these patients, a near infrared spectroscopy-based BCI has been developed, allowing them to reply to “yes”/”no” questions with a classification accuracy of 70%. Because of the non-optimal accuracy, a usual character-based speller for selecting letters or words cannot be used. In this paper, a novel spelling interface based on the popular 20-questions-game has been presented, which will allow patients to communicate using only “yes”/”no” answers, even in the presence of poor classification accuracy. The communication system is based on an artificial neural network (ANN) that estimates a statement thought by the patient asking less than 20 questions. The ANN has been tested in a web-based version with healthy participants and in offline simulations. Both results indicate that the proposed system can estimate a patient’s imagined sentence with an accuracy that varies from 40%, in the case of a “yes”/”no” classification accuracy of 70%, and up to 100% in the best case. These results show that the proposed spelling interface could allow patients in CLIS to express their own thoughts, instead of only answer to “yes”/”no” questions.

## 1. Introduction

In the past decades, many alternative communication systems have been developed for people with speech, language, or motor impairments. Brain computer interfaces (BCI) were developed to provide a means of communication for people with severe motor disabilities (for review see Chaudhary et al., 2016) [1,2,3]. The most commonly used non-invasive BCI spelling application is based on the electroencephalography (EEG) based P300 event-related brain potential, where a patient can select letters from a matrix in which each character is transiently illuminated [4]. Another BCI system commonly used to select letters from a screen is based on steady state visually evoked potential (SSEVP) [5,6]. Other BCI communication systems are based on slow cortical potential [7], and on the sensorimotor rhythm of the EEG [8,9] to control cursors or keyboards on a screen. These systems, even using different signals and different interfaces, are all based on the same general paradigm, namely, that patients communicate by selecting letters or words from a screen. Different features and classification techniques are used to decode the intention of patients [10,11,12]. Independently from the signal type, all of these BCI systems are based on the control of a neuroelectric brain response, and the learning process is based on feedback and reward. Despite the good results achievable using these systems with patients suffering from disorders leading to loss of communication, none of these techniques were able to provide a means of communication to amyotrophic lateral sclerosis (ALS) patients in a completely locked-in state (CLIS). An explanation of the non-applicability of the standard BCI in complete paralysis with otherwise intact cognitive processing, Kübler and Birbaumer suggested the theoretical psychophysiological notion of “extinction of goal directed cognition and thought” in CLIS [13]. Following this idea, a BCI based on functional near-infrared spectroscopy (*f*NIRS) was developed for ‘reflexive’ communication in CLIS. Unlike the other communication systems, it allows the patient to answer short questions affirmatively (“yes”) and negatively (“no”), using the blood oxygenation change of their fronto-central brain regions. The best accuracy reported for correctly classified “yes”/”no” answers is 70% in CLIS [14,15]. The low classification accuracy and the only binary “yes”/”no” answers do not allow the patients to express their own thoughts using a classic character-selection-based speller, but only to answer prerecorded questions.

The limitations of the *f*NIRS-BCI, especially the restriction to a binary “yes”/”no” signal and a substantial error rate, are common not only to all non-invasive BCI systems, but also to all the telecommunication systems. Using telecommunication words, the BCI problem involves the correct detection of a communication between two agents through a noisy channel. The communication, both in the general case of telecommunication or in the particular case of the “yes”/”no”-BCI, is a binary message sent from the sender (or the brain) to the receiver (the computer), whose information may be distorted in the transmission due to the noise in the channel (wrong classification), and the task of the receiver is to recover the message reconstructing the corrupted signal [16].

The BCI-spellers usually solve the problem of the wrong signal classification with a redundant number of inputs (e.g., flashing each letter multiple times in order to be sure that the selection was not due to a false positive). With the *f*NIRS-BCI, this technique is because of the characteristic of the *f*NIRS signal; the *f*NIRS-BCI system is slow and allows the patient to answer approximately only one question every 20 s. The solution for this kind of BCI would be a speller capable of correcting the errors in the classification of the answers, allowing a patient to communicate using minimum number of inputs.

A solution can be found in a popular game, the 20-questions-game. In this game, a player has to guess what the other player is thinking within 20 “yes”/”no” questions. An electronic version of the game, which has been played more than 88 millions times, can correctly guess what someone is thinking with 80% precision, by asking 20 questions (95% of the time with 25 questions) [17]. The game was mathematically formalized by Alfred Rényi [18] and it was later proposed in a different version by Stanisław Ulam [19]. The Rényi¬–Ulam game and its variations have been used to solve many different problems [20,21,22], in this paper we propose to use the game as a spelling interface for a binary BCI, like the *f*NIRS-based BCI described in Chaudhary et al. (2017). This kind of communication system may allow patients in CLIS to express their own thoughts and not just to reply to prerecorded questions.

The rest of the paper is structured as follows: in Section 2, the method used to design the communication system is described, and in particular, in Section 2.1 and Section 2.2 describe the algorithm of the Rényi–Ulam game and its application to the popular 20-questions-game using an artificial neural network, and in Section 2.3, the implementation as an interface for a BCI system is described. In Section 3, the proposed algorithm is explained in detail. Then, in Section 4, we present the results of the algorithm, both for an online version of the game played by real persons (Section 4.1) and for an offline version with computer simulations (Section 4.2). The results are discussed and followed by the conclusion in Section 5. While the databases used for the results are described in Appendix A.

## 2. Materials and Methods

### 2.1. Rényi–Ulam Game

The 20-questions-game is a popular game played by two players. The rules of the game are as follows: the first player (player *A*, the Responder) imagines a famous person, while the second (player *B*, the Questioner) must guess the person by asking twenty “yes”/”no” questions (e.g., “Is the person alive?”).

The game has been mathematically described by Rényi and Ulam, as follows: the Responder can imagine any target statement that is contained in a fixed search space (i.e., the topic, e.g., famous people), while the Questioner has to guess the statement using less than *n* (e.g., 20) “yes”/”no” questions. Moreover, the Responder is allowed to lie up to *e* times on the answers given to the “yes”/”no” questions (i.e., they can give wrong answers). The lies are a formalization of the wrong answer that a player can give if their knowledge about the statement is different from the knowledge of the other player (e.g., the Responder thinks that a person is alive, but instead it is dead).

The complete description of the game is outlined below:The game is played by two players: *A* (the Responder) and *B* (the Questioner).A set *S* of target statements (the search space) is fixed.A number *n* > 0 of questions is fixed.An upper bound *e* ≥ 0 of number of lies is fixed.*B* can ask questions in the form of “Is *x* in *T*?”, where *T* is a subset of *S.**A* must reply “yes” or “no”, and he can lie up to *e* times.*B* wins if he can correctly guess *x* after *n* questions.

The number of questions *n* to solve the Rényi–Ulam game depends linearly on the cardinality of *S* and on the maximum number of lies *e*, but for the general case of an arbitrary number of lies, there is no general solution and only heuristic methods have been proposed [23].

### 2.2. Artificial Neural Network

A heuristic solution of the Rényi–Ulam game with arbitrary number of lies can be found using an artificial neural network (ANN). This method was first developed by Robin Burgener [24] for *20q*, an electronic version of the 20-question-game. This version is slightly different from the Rényi–Ulam game; for instance, the allowed answers are not only “yes” and “no”, but also “unknown”, “irrelevant”, “sometimes”, “depends”, etc. Here, we propose an ANN for the original Rényi–Ulam game with binary answers only.

The ANN will play the role of the Questioner, that is, it will ask questions, and it will estimate a particular target statement (e.g., a person) imagined by a Responder. Therefore, in order to work, the ANN needs two databases, one with the target statements belonging to the search space (e.g., all of the possible famous people), and one with the possible “yes”/”no” questions (e.g., “Is it alive?”, “Is it a woman?”, etc.). 

The main core of the ANN is the relation between the statements and questions. Each target statement is connected to each question, and the strength of this connection is indicated by a weight. The weights can be negative if the statement and question are not related (i.e., the expected answer is “no”) and positive if they are related (i.e., the expected answer is “yes”). All of the weights are stored in a matrix called a weight matrix.

The ANN will present to the Responder the questions stored in the database. The choice of the question is based on the weight table and on the previous questions.

The final estimation of the ANN is the statement that, based on the received answers, is the most probable. In order to calculate this probability, after each question, the ANN will penalize or reward, based on the answer, the target statements (e.g., if the answer to “Is she a woman?” is “yes”, all male persons will be penalized).

Finally, after each correct final estimation, the weight matrix is updated based on the received answer, allowing a learning process.

Using ANN has two advantages. First, if the Responder occasionally lies, the ANN will not exclude any possible target statement, based on that single answer, but it will only change the probability for the final estimation. Second, the estimation of the target statement will improve with frequent usage of ANN, because the learning process improves the reliability of the weight table.

### 2.3. 20-Questions-Based Interface for Communication Systems

#### 2.3.1. Proposed BCI Implementation

We endeavor to use the 20-questions-game as a communication system for patients that do not have a reliable means of communication, like patients in a complete locked-in state (CLIS). This system is based on an ANN that interacts with the patient in a 20-questions-based paradigm, in order to estimate their thoughts. 

For this purpose, the ANN can be developed as part of a brain-computer interface; the computer proposes auditorily the questions to the patient, and it records a brain signal (e.g., *f*NIRS). The BCI classifies the brain signal in a binary answer (“yes” or “no”), which will be the answer required by the ANN. In this implementation, the patient will play the role of the Responder, while the ANN will be the Questioner. The patient can think of any word or sentence that is stored in the database of the ANN, and the ANN will ask questions, also stored in the database, in order to estimate the patient’s thought. The “yes”/“no” classification accuracy achieved using BCI systems with CLIS patients is around 70% [14,15]. Using the 20-questions-based system, the errors on the “yes”/“no” classification will be considered as the lies of the Rényi–Ulam game, therefore, they will not automatically lead to a wrong estimation of the sentence.

The proposed 20-questions-based communication system is depicted in Figure 1. The system has been tested as a communication system, independently from the brain signal records, with healthy participants, using a web interface, and with computer simulations.

#### 2.3.2. Web-Based Implementation

The web-based version of the algorithm (www.alsbci.eu) was written in Python and it has been translated into three languages, English, German, and Italian.

In the website, the user is asked to put himself in a complete locked-in patient’s shoes, playing the 20-questions-game by thinking a sentence that could be asked by a patient in such conditions. The search space was intentionally left ambiguous and not bound to a specific topic, in order to check the performance of the system in a not optimal scenario. The user had also the option to check the list of target statements already stored in the database.

During the game, the ANN presented the questions to the user, who had the opportunity to reply “yes”, “no”, or “unsure”. In the case of an “unsure” answer, the ANN ignored the answer and, instead, it was asking a different question. At the end of each game, the ANN tried to estimate the thought sentence three times, proposing to the users the three most probable targets (i.e., the three statements with the highest current value). Finally, if none of the proposed target statements was the correct one, the user could select (or, if not present, insert) the thought sentence directly from the database.

From the website, the users had also the opportunity to improve the databases of the ANN by adding new statements and questions.

The web-based version was initialized with an initial database manually populated with a set of 41 target statements and 25 questions. The website has been online, accessible to everyone since November 2017. Since then, the game has been played 92 times, and 50 new statements and 113 new questions have been added to the system, bringing the total number to 91 statements and 138 questions, respectively (see Appendix A).

#### 2.3.3. Simulation

Using an offline version of the website, we tested the algorithm by changing the possible answers and simulating a BCI with errors on the classification of the “yes” and “no” answers.

Regarding the possible answers, we considered three different cases, as follows: “yes”, “no”, and “unsure” answers, with the questions answered as “unsure” excluded from the total number of questions (same as the online system);“yes”, “no”, and “unsure” answers, with the questions answered as “unsure” included in the total number of questions; and“yes” and “no” answers only.

As the expected answer is a direct expression of the target-question weight, we considered a “yes” answer when the weight was positive, “no” when negative, and “unsure” when the weight was zero. In the third case, considering the “yes” and “no” answers only, if the target-question weight was zero, we chose “yes” or “no” randomly.

In order to emulate the non-optimal BCI classification, according to the simulated accuracy, each answer had a certain probability of being wrong (if “unsure”, the answer was not changed). The algorithm performance has been tested, varying the classification accuracy between 50% (i.e., random classification) and 100% (i.e., perfect classification). As for the online and the offline analyses, we considered a statement as correctly estimated if, after 20 questions, it was among the three most probable target statements.

## 3. Algorithm

### 3.1. Definitions

The two main agents of the ANN are the target statements (i.e., the possible final sentences) and the questions (i.e., the descriptors of the sentences). Both of the target statements and sentences are stored in a database, therefore, the only possible sentences and questions are the ones present in the communication system.

As explained in Section 2.2, the core of the ANN is the weight matrix that puts in relation the target statements and questions. The weight depends on the answer that each statement is required from each question (i.e., if the expected answer is “yes”, the weight will be positive, if “no”, it will be negative).

A value is assigned to each statement. This value indicates the probability of each statement to be the final target; the higher the value assigned to one statement, the higher the probability of that statement to be the thought one. The value is updated after each question, based on the statement–question weight and on the received answer.

The elements of the ANN are shown in Figure 2, and are summarized below:*N* targets (*T_i_* with *i =* 1:*N*) (i.e., sentences thought by the patient);Each target is described by *M* descriptors (*D_j_* with *j =* 1:*M*) (i.e., “yes”/”no” questions);Strength of *T­–D* connection is expressed by a weight (*W_Ti,Dj_* with *i =* 1:*N, j =* 1:*M*); andEach target *T_i_* is ranked using a current value (*V_Ti_* with *i =* 1:*N*).

### 3.2. Current Value Adjustment

During each run, all of the target statements start with the same probability of being the final sentence, therefore, all of the current values $V_{T}$ are initialized to 0. This probability (i.e., the current value) changes after each presented question, based on the answer of the user. In particular, if $D_{j}$ is the *n*-th question presented to the user, for each target statement $T_{i}$, the current value $V_{T_{i}}$ is updated using the formula, as follows:$$\;V_{T_{i}}\left(n\right)=\;V_{T_{i}}\left(n-1\right)+W_{T_{i},D_{j}}\;\mathrm{if}\;\mathrm{answer}\;\mathrm{is}\;“\mathrm{yes}”\;$$
$$\;V_{T_{i}}\left(n\right)=\;V_{T_{i}}\left(n-1\right)-W_{T_{i},\;D_{j}}\;\mathrm{if}\;\mathrm{answer}\;\mathrm{is}\;“\mathrm{no}”$$
where n is the number of the question, and $W_{T_{i},D_{j}}$ is the weight between question $D_{j}$ and statement $T_{i}$. It is positive if the expected answer is “yes” and negative if the expected answer is “no”. Therefore, the formula increases the current value if the given answer is the expected one, and decreases it otherwise.

In order to decrease the impact of the wrong answers, the adjustment of the current value has been increased for those statements that receive many answers coherent with the expected ones. After each question, every statement where the expected answer matches with the received one is marked as a ‘priority target’. This priority is lost whenever the statement receives an answer that does not match with the expected answer. The priority targets receive an adjustment for their current value, equal to double the weight. This leads to the following modified formula for updating the current value:$$\;V_{T_{i}}\left(n\right)=\;V_{T_{i}}\left(n-1\right)+W_{T_{i},D_{j}}(\times 2\;\mathrm{if}\;T_{i}\;\mathrm{has}\;\mathrm{priority})\;\mathrm{if}\;\mathrm{answer}\;\mathrm{is}\;“\mathrm{yes}”$$
$$\;V_{T_{i}}\left(n\right)=\;V_{T_{i}}\left(n-1\right)-W_{T_{i},\;D_{j}}(\times 2\;\mathrm{if}\;T_{i}\;\mathrm{has}\;\mathrm{priority})\;\mathrm{if}\;\mathrm{answer}\;\mathrm{is}\;“\mathrm{no}”$$
where the variables are the same as described above.

### 3.3. Choice of the Question

One of the crucial points of the algorithm is the choice of the question. The best question is the one whose answer will give more information about the most probable targets, or, in other words, the one whose answer splits the most probable targets in two similar sets. Therefore, the best question is the one that maximizes the entropy
$$\;H\left(D_{j}\right)=\;\underset{x\in X}{\sum }-p\left(x\right)\log_{2}p\left(x\right)$$
where *X* is the two classes of statements with positive and negative weights, with respect to the question *D_j_*; and *p*(*x*) is the proportion of the most probable statements that belong to the class *x*.

In the implementation, all of the targets with a positive current value were considered as the most probable targets. It is possible to choose the most probable targets in a different way, using a more or less strict definition (e.g., the targets with a current value greater than a certain threshold), and this will obviously change the choice of the questions accordingly.

### 3.4. Estimate the Target

The goal of the ANN is to estimate the target statement that the patient is thinking. After 15 questions, the ANN will check if there is only one target statement with a positive value; if this happens, it will estimate that statement. If this condition never occurs, after 20 questions, the ANN will estimate the target statement with the highest current value.

The lower threshold of 15 questions is based on the minimum number of questions needed for an optimal solution of the Rényi–Ulam game; considering a search space of 91 statements and a signal classification accuracy of 75%, the minimum number of questions for a deterministic optimal solution is 23 (Table 2.3 from Cicalese, 2013, p. 28). We decided to check whether there was only one statement with a positive value after two thirds of the minimum number of questions for an optimal solution. This condition is meant to speed up the communication process, avoiding asking unnecessary questions when one statement is likely the correct target.

### 3.5. Learning Step

The last step of the algorithm is teaching the neural network. After each correct estimation, the system will update the weight matrix. For each question that was asked during the run, it will update the weight that associates that question to the correctly estimated statement, based on the answer that the user gave; if the given answer is “yes”, it will increase the weight value, otherwise it will decrease it. In order to avoid excessive values, the weights are upper and lower bounded.

## 4. Results

In the next paragraphs the online and offline results of the proposed algorithm will be presented. The results are based on the web-based version and on the simulations descripted in Section 2.3.2 and Section 2.3.3, respectively.

### 4.1. Online Results

The results of the games played online are reported in Table 1. Half of the time the game was played with a statement that was not in the system; considering that only the games that played with statements already in the system, the percentage of correct estimations is 65.95%, against 34.04% of games where the ANN was not able to correctly estimate the thought sentence. Focusing on the sentences correctly estimated, 67.74% of the time the sentence was estimated on the first attempt.

### 4.2. Offline Results

The offline results, reported in Figure 3, were obtained by simulating the performance of the ANN in the cases mentioned in Section 2.3.3. For each of the three cases, the simulation was performed by varying the signal classification accuracy between random (i.e., 50%) and perfect (i.e., 100%). Figure 3a–c represents the percentage of statements correctly estimated by the ANN after 1000 simulations, with respect to the simulated BCI classification accuracy of “yes” and “no”. In each figure, blue, green, and yellow represent the percentage of statements correctly estimated as the most, second most, and third most probable statement, respectively.

In order to evaluate the time performance of the proposed communication system, we compared the typing speed of the ANN to those of the classic P300-based matrix speller [25]. The *f*NIRS-based BCI developed for CLIS patients is able to present one question every 20 s [15]. Therefore, a spelling interface that uses this BCI has an information transfer rate (ITR) of 3 bits/min, while the matrix speller reaches 12 bits/min, which means a typing speed of approximately one character every 26 s. The target statements in the database of the ANN (Table A1) have an average length of 23.625 characters. Hence, as in the simulations, the statements were estimated in 20 questions, the *f*NIRS-BCI for the CLIS patients using the 20-questions-based spelling interface will have an average typing speed of one character every 17 s.

## 5. Discussion and Conclusions

The results in the offline analyses show that the performances are very similar in the first two analyzed cases, discarding and including “unsure” answers. Surprisingly, when giving random answers instead of “unsure”, the results improve. We believe that this is due to the randomization of the target statements and does not represent a real improvement in the results.

Figure 3a–c shows that considering a classification accuracy of 100%, the ANN is always able to correctly estimate the target statement. This result means that, using a BCI that perfectly classifies “yes” and “no” answers, a patient could communicate entire words, or even sentences, by answering only 20 questions. The result is very promising, considering that, under the condition of a perfect signal classification, in order to select one character, a usual 6 × 6 grid-based speller needs at least 12 inputs [26].

However, we also notice that if the accuracy drops down to 80%, the correct rate decreases to 57%. Nevertheless, we have to consider that we did not put any constraint on the possible target statements, so in the same database, there were very different sentences like, “This movie is beautiful” and “I would like to go more out from the bed”. This generality of the sentences put the program in a bad case scenario. Although, it is important to notice that these results are still significant, as, considering a random classification (accuracy of 50%), the correct rate is close to 0%.

Both in the online games and in the simulations, the system always asked 20 questions, therefore, after 15 questions, there were always at least two statements with positive value. Hence, the ANN always estimated the final target statement with a certain degree of uncertainty, probably because the number of played games was not enough for an optimal training of the weight table. In order to decrease the uncertainty, a possibility is to increase the number of questions from 20 to the optimal solution number, which depends on the cardinality of the search space and on the signal classification accuracy, as shown in Table 2.3, from Cicalese, 2013, p. 28. Nonetheless, we decided to keep the upper limit of 20 questions in order to build a communication system that could be used in a reasonable time, even using a *f*NIRS-based BCI (20 s for each question).

The comparison between the 20-questions-based system and the P300 matrix speller shows that, despite a lower ITR, the average typing speed of the proposed spelling interface is higher. Even if this result cannot be taken as a real typing speed comparison because the ANN can estimate only entire sentences, it shows that the proposed system has time performance comparable to the usual spellers and could allow communication in a reasonable time, even in presence of a slow signal like the *f*NIRS (3 bits/min).

Correlating the online and the offline results, we can say that the users gave the expected answers up to 85% of the time. Obviously, in that case, there were no errors in the signal classification, but we could not expect a perfect result because the questions could have been very general, and with a not unique answer (e.g., considering the sentence “I sleep a lot”, the question “Is it positive?” could be answered “yes” or “no” depending on the positive or negative connotation that a person gives to sleeping a lot).

The results show that the 20-questions-based system can be a valid interface for any BCI that uses a slow signal and/or has a classification with a low accuracy rate. Even in presence of fast signal (e.g., EEG), the proposed system can improve the typing speed performance, allowing the formulation of entire sentences using only 20 binary inputs. The main drawback, already highlighted in the previous sections, is that the only sentences that the ANN can estimate are the ones stored in the database, therefore, a patient will not be free to formulate his own sentences. This limitation, an intrinsic characteristic of a 20-questions-system, can be overcome by building an exhaustive database personalized for each patient. Before initiating any BCI session, the patients will be provided an option to choose between the proposed 20-questions-based system and a character-selection speller that gives more freedom at the expense of the typing speed and the error handling.

In the future, we will test the system by narrowing the possible sentences to a more restricted topic and personalizing the weight table for only one person, in order to adapt the weights to his or her individual biography and personality. Moreover, the system will be improved to work with multi-class BCIs, in order to have more possible answers and, therefore, better estimations. Finally, the interface will be tested with a BCI to study the reaction of the patients to this different approach of communication.

The results are promising and show that a communication system based on this algorithm could replace the usual speller-based approach. The main limitation of the 20-questions-based interface is that it does not allow the patient to create new sentences or new questions. Nevertheless, it could allow patients in CLIS to express their own thoughts and desires, instead of only answering to “yes”/”no” questions chosen by someone else. For this reason, the communication system based on the proposed algorithm could be applied to estimate the inner mental and thought process of patients in CLIS.

## Figures and Tables

**Figure 1 brainsci-08-00126-f001:**
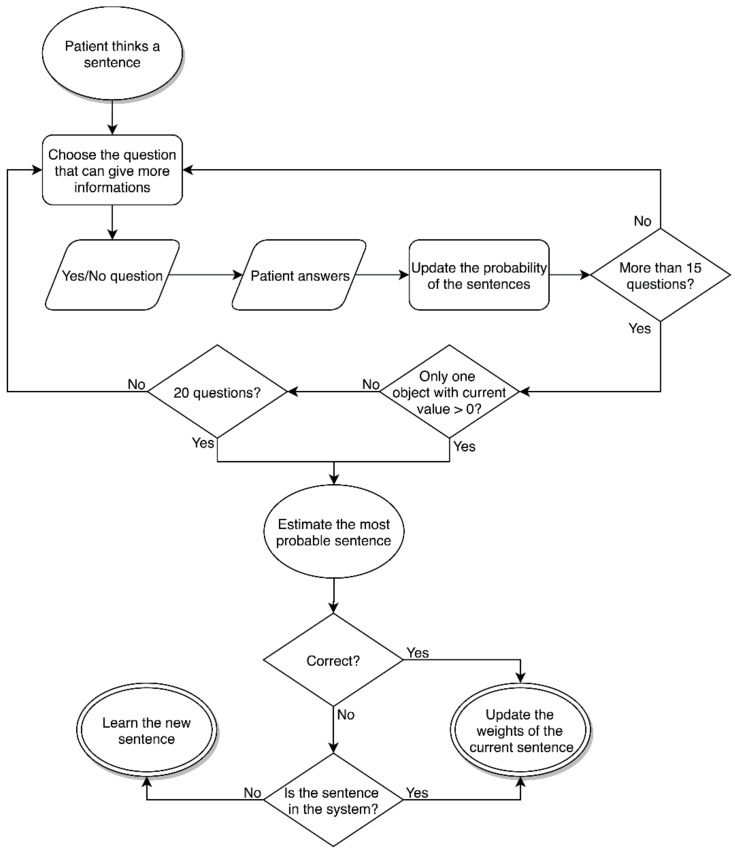
Flow chart of the proposed 20-questions-based communication system.

**Figure 2 brainsci-08-00126-f002:**
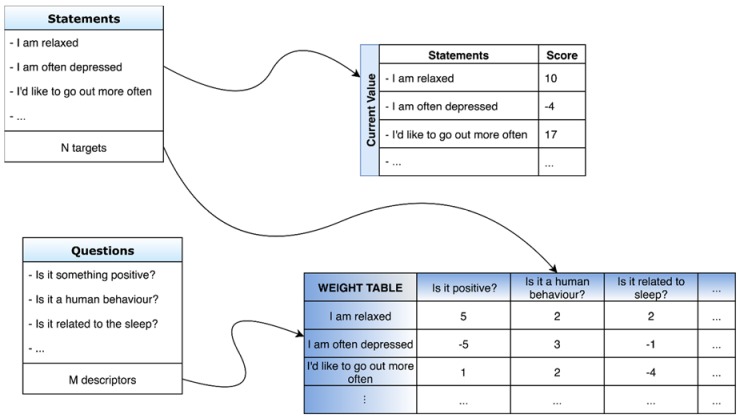
Structure of the artificial neural network. In particular, the structure of the databases of statements and questions, of the table of current values, and of the weight table are shown.

**Figure 3 brainsci-08-00126-f003:**
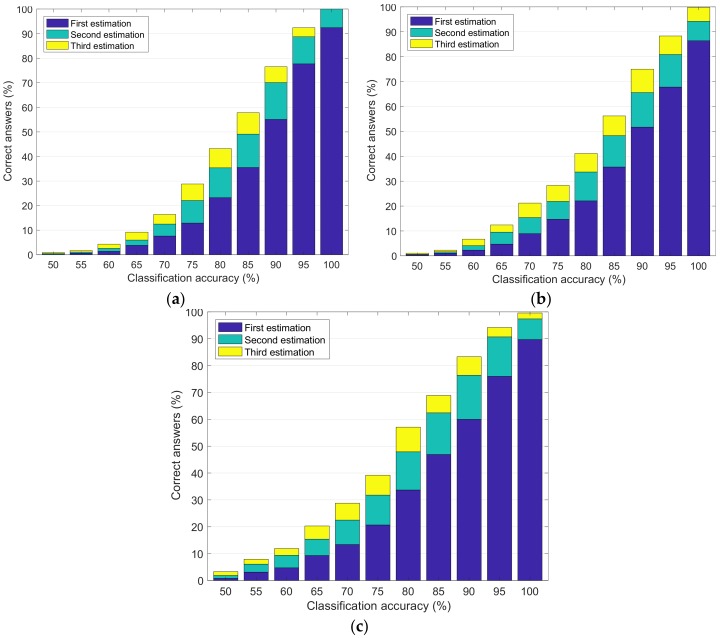
Results of the offline simulation in the three different cases. Blue, green, and yellow represent the percentage of statements correctly estimated as most, second most, and third most probable statement, respectively. (**a**) Simulated results using “yes”, “no”, and “unsure” answers, with the questions answered as “unsure” excluded from the total number of questions; (**b**) simulated results using “yes”, “no”, and *“*unsure” answers, with the questions answered as “unsure” included in the total number of questions; and (**c**) simulated results using “yes” and “no” answers only.

**Table 1 brainsci-08-00126-t001:** Results of the game played online on the website. The table lists the total number of times of the game play. The game was played for a total of 92 times, out of which it was played for 45 times on new statements (not in the database) and 47 times on old statements (in the database). For the statements already in the database, the table also lists the number of times that they were estimated incorrectly and correctly. For the correctly estimated statements the table lists the number of times the statements were the first, second, or third guess.

New Statements	Old Statements			
**45**	47			
	Incorrect	Correct		
	16	31		
		1st Estimation	2nd Estimation	3rd Estimation
		21	5	5

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
