# Peer review of "A 20-Questions-Based Binary Spelling Interface for Communication Systems"

_brainsci, 2018, doi:10.3390/brainsci8070126_

Round 1

Reviewer 1 Report

The authors introduced a novel spelling interface based on the popular 20-questions-game has been presented, that will allow patients to communicate using only yes/no answers, even in the presence of poor classification accuracy. Experimental study was implemented to test the performance of this system. Overall the paper is well written and the results are convincing. The following minor concerns need to be addressed before acceptance for publication: 

1. An overall flowchart of the designed system could be helpful for the reader to better understanding the proposed method.

2. In a separate paragraph it is required to provide some including remarks to further discuss the proposed methods, for example, what are the main advantages and limitations in comparison with existing methods?

3. There are many relevant studies in terms of both feature selection and classifier training for EEG classification in BCI applications, but were not mentioned in this paper. The authors need to further elaborate the introduction or review of the existing literatures, for example: Discriminative feature extraction via multivariate linear regression for SSVEP-based BCI; Sparse Bayesian classification of EEG for brain-computer interface; Sparse group representation model for motor imagery EEG classification; Multi-kernel extreme learning machine for EEG classification in brain-computer interface; A novel multilayer correlation maximization model for improving CCA-based frequency recognition in SSVEP brain-computer interface.

4. What are future endeavors of your study? Please open a real window for future work in the conclusion section.

Author Response

The authors introduced a novel spelling interface based on the popular 20-questions-game has been presented, that will allow patients to communicate using only yes/no answers, even in the presence of poor classification accuracy. Experimental study was implemented to test the performance of this system. Overall the paper is well written and the results are convincing. The following minor concerns need to be addressed before acceptance for publication:

Response: Thank you for your comments, they are highly valuable, we have changed our manuscript as per your suggestions as shown below.

1. An overall flowchart of the designed system could be helpful for the reader to better understanding the proposed method.

Response: Authors thanks the reviewer for the comment, but we would like to mention that the Figure 1 of original manuscript explicates the flow chart of the proposed system. We are not sure on which aspect of the flowchart we need to elaborate, could the reviewer specify the changes we needed to done in Figure 1?

2. In a separate paragraph it is required to provide some including remarks to further discuss the proposed methods, for example, what are the main advantages and limitations in comparison with existing methods?

Response: Thank you for the comment this was very help. We have written a new paragraph on advantages and limitations in the updated manuscript from line no. 423-432.

Changes in Manuscript:

“The results show that the 20-questions-based system can be a valid interface for any BCI that uses a slow signal and/or have a classification with low accuracy rate. Even in presence of fast signal (e.g. EEG) the proposed system can improve the typing speed performance allowing the formulation of entire sentences using only 20 binary inputs. The main drawback, already highlighted in the previous sections, is that the only sentences that the ANN can estimate are the ones stored in the database, therefore a patient will not be free to formulate his own sentences. This limitation, intrinsic characteristic of a 20-questions-system, can be overcome by building an exhaustive database personalized for each patient. Before initiating any BCI session, the patients will be provided an option to choose between the proposed 20-questions-based system and a character-selection speller that gives more freedom at the expense of the typing speed and the error handling.”

3. There are many relevant studies in terms of both feature selection and classifier training for EEG classification in BCI applications, but were not mentioned in this paper. The authors need to further elaborate the introduction or review of the existing literatures, for example: Discriminative feature extraction via multivariate linear regression for SSVEP-based BCI; Sparse Bayesian classification of EEG for brain-computer interface; Sparse group representation model for motor imagery EEG classification; Multi-kernel extreme learning machine for EEG classification in brain-computer interface; A novel multilayer correlation maximization model for improving CCA-based frequency recognition in SSVEP brain-computer interface.

Response: Authors thanks the reviewer for the comment. We have added the appropriate reference at their respective places and added sentences on SSVEP and feature selection in BCI from line no. 33 - 39.

Changes in Manuscript:

Included references and updated the reference list.

6.      Jiao, Y.; Zhang, Y.; Wang, Y.; Wang, B.; Jin, J.; Wang, X. A novel multilayer correlation maximization model for improving CCA-based frequency recognition in SSVEP brain--computer interface. Int. J. Neural Syst. 2018, 28, 1750039.

10.   Zhang, Y.; Zhou, G.; Jin, J.; Zhao, Q.; Wang, X.; Cichocki, A. Sparse Bayesian Classification of EEG for Brain-Computer Interface. IEEE Trans. Neural Networks Learn. Syst. 2015, doi:10.1109/TNNLS.2015.2476656.

11.   Jiao, Y.; Zhang, Y.; Chen, X.; Yin, E.; Jin, J.; Wang, X. Y.; Cichocki, A. Sparse Group Representation Model for Motor Imagery EEG Classification. IEEE J. Biomed. Heal. Informatics 2018, doi:10.1109/JBHI.2018.2832538.

12.   Zhang, Y.; Wang, Y.; Zhou, G.; Jin, J.; Wang, B.; Wang, X.; Cichocki, A. Multi-kernel extreme learning machine for EEG classification in brain-computer interfaces. Expert Syst. Appl. 2018, doi:10.1016/j.eswa.2017.12.015.

“Another BCI system commonly used to select letters from a screen is based on steady state visually evoked potential (SSEVP) [5, 6]. Other BCI communication systems are based on slow cortical potential [7], and on the sensorimotor rhythm of the EEG [8, 9] to control cursors or keyboards on a screen. These systems, even using different signals and different interfaces, are all based on the same general paradigm: patients communicate by selecting letters or words from a screen. Different features and classification techniques are used to decode the intention of patients [10-12].”

4. What are future endeavors of your study? Please open a real window for future work in the conclusion section.

Response: Thank you for the comment. In our original manuscript we wrote following as the last paragraph of our paper

The results are promising and show that a communication system based on this algorithm could replace the usual speller-based approach. The main limitation of the 20-questions-based interface is that does not allow the patient to create new sentences or new questions. Nevertheless, it could allow patients in CLIS to express their own thoughts and desires, instead of only answer to yes/no questions chosen by someone else. For this reason, the communication system based on the proposed algorithm will be applied to estimate the inner mental and thought process of patients in CLIS.”

This paragraph highlights the real application of our proposed method, estimating and investigating he inner thought process of patients without means of communication.

In addition to this, to address the comment of the reviewer we wrote followings lines in our updated manuscript from line no. 435 – 438.

Changes in Manuscript:

“Moreover, the system will be improved to work with multi‑class BCIs, in order to have more possible answers and, therefore, better estimations. Finally, the interface will be tested with a BCI to study the reaction of the patients to this different approach on the communication.”

Reviewer 2 Report

This is a very interesting article with both theoretical innovation and practical value. The method developed in this study is not only useful for BCI but may also have a broader impact in the relevant fields. I have a few suggestions which may help authors further improve the manuscript.

1.       For describing the algorithm, it is better to use “estimate” instead of “guess”. “Guess” is very subjective, which is more suitable for human beings rather than a machine/algorithm.

2.       Introduction: BCI based on the slow cortical potential or sensorimotor rhythm are mainly used for movement control instead of selecting letters. Selecting letters are often realized by P300 or SSVEP BCI.  Ref [9] is a sensorimotor rhythm based BCI instead of a slow cortical potential based BCI. Authors may refer to “Subject-specific time-frequency selection for multi-class motor imagery-based BCIs using few Laplacian EEG channels. Biomedical Signal Processing and Control, 38, pp.302-311, 2017.” to get a general idea of sensorimotor rhythm based BCI and how it different from other BCI.

3.       Section 2.2 the sentence “Second, the learning process entails that the more the ANN is used, the more the weight table will be reliable, and the guesses become more accurate” has syntax problems and needs to be rephrased.

4.       The results in Fig 3 showed that the lies of the Renyi-Ulam game (classification accuracy) did affect the ANN estimation (correct answer percentage). However, in the end of 2.3.1, the authors argued that the lies of the Renyi-Ulam game will not have a dramatic impact on the final guess o the ANN. Please correct this misleading statement.

5.       The first paragraph of Section 4.1 is part of method. Therefore, it is better to move this paragraph to a Method section.

6.       Why first use 15 question? Does this number is optimal for the first attempt? How to determine this number? Does it depend on the total number of questions?

7.       I think the optimal number of total questions should be depended on the lie rate of the Renyi-Ulam game and the target space. Please provide a discussion on this issue.

Author Response

This is a very interesting article with both theoretical innovation and practical value. The method developed in this study is not only useful for BCI but may also have a broader impact in the relevant fields. I have a few suggestions which may help authors further improve the manuscript.

Response: Thank you for your comments, they are highly valuable, we have changed our manuscript as per your suggestions as shown below.

1.       For describing the algorithm, it is better to use “estimate” instead of “guess”. “Guess” is very subjective, which is more suitable for human beings rather than a machine/algorithm.

Response: Thank you for the comment this was very help. We have the updated the manuscript by replacing the word “guess” with “estimate” throughout the manuscript.

Changes in Manuscript:

The word “guess” has been replaced with “estimate” throughout the manuscript, included figures and table

2.       Introduction: BCI based on the slow cortical potential or sensorimotor rhythm are mainly used for movement control instead of selecting letters. Selecting letters are often realized by P300 or SSVEP BCI.  Ref [9] is a sensorimotor rhythm based BCI instead of a slow cortical potential based BCI. Authors may refer to “Subject-specific time-frequency selection for multi-class motor imagery-based BCIs using few Laplacian EEG channels. Biomedical Signal Processing and Control, 38, pp.302-311, 2017.” to get a general idea of sensorimotor rhythm based BCI and how it different from other BCI.

Response: Authors thanks the reviewer for the comment and sharing the knowledge with us. We would like to apologize for the mistake with the references; we have updated the reference list and numbering. We also added sentences in the introduction in response to comment 3 of reviewer 1. Please see the response to comment 3 of reviewer 1.

Changes in Manuscript:

9.      Yang, Y.; Chevallier, S.; Wiart, J.; Bloch, I. Subject-specific time-frequency selection for multi-class motor imagery-based BCIs using few Laplacian EEG channels. Biomed. Signal Process. Control 2017, doi:10.1016/j.bspc.2017.06.016.

3.       Section 2.2 the sentence “Second, the learning process entails that the more the ANN is used, the more the weight table will be reliable, and the guesses become more accurate” has syntax problems and needs to be rephrased.

Response: Authors apologize for the unclear statement. We changed the sentence in the updated manuscript at line no. 158-159.

Changes in Manuscript:

“Second, the estimation of the target statement will improve with frequent usage of ANN because the learning process improves the reliability of the weight table.”

4.       The results in Fig 3 showed that the lies of the Renyi-Ulam game (classification accuracy) did affect the ANN estimation (correct answer percentage). However, in the end of 2.3.1, the authors argued that the lies of the Renyi-Ulam game will not have a dramatic impact on the final guess o the ANN. Please correct this misleading statement.

Response: Authors thanks the reviewer for the comment and for pointing out the fallacy. We changed the sentence in the updated manuscript from line no. 184 -186.

Changes in Manuscript:

“Using the 20-questions-based system, the errors on the “yes”/“no” classification will be considered as the lies of the Rényi-Ulam game, therefore they will not automatically lead to a wrong estimation of the sentence.”

5.       The first paragraph of Section 4.1 is part of method. Therefore, it is better to move this paragraph to a Method section.

Response: Thank you for your comment, we moved the paragraph as suggested.

Changes in Manuscript:

Moved the first paragraph of Section 4.1 and placed it at the end of section 2.3.2

6.       Why first use 15 question? Does this number is optimal for the first attempt? How to determine this number? Does it depend on the total number of questions?

Response: Authors thanks the reviewer for this valuable comment, we have added details on these in the updated manuscript from line no. 298-302.

Changes in Manuscript:

“The lower threshold of 15 questions is based on the minimum number of questions needed for an optimal solution of the Rényi-Ulam game: considering a search space of 91 statements and a signal classification accuracy of 75% the minimum number of questions for a deterministic optimal solution is 23 (Table 2.3 from Cicalese, 2013, p. 28). We decided to check if there was only one statement with positive value after two third of the minimum number of questions for an optimal solution.”

7.       I think the optimal number of total questions should be depended on the lie rate of the Renyi-Ulam game and the target space. Please provide a discussion on this issue.

Response: Authors thanks the reviewer for the comment, we added a discussion on this issue in the discussion section of the updated manuscript from line no. 395 – 403.

Changes in Manuscript:

“Both in the online games and in the simulations the system always asked 20 questions, therefore after 15 questions there were always at least two statements with positive value. Hence, the ANN always estimated the final target statement with a certain degree of uncertainty, probably because the number of played games was not enough for an optimal training of the weight table. In order to decrease the uncertainty, a possibility is to increase the number of questions from 20 to the optimal solution number, that depends on the cardinality of the search space and on the signal classification accuracy as shown in Table 2.3 from Cicalese, 2013, p. 28. Nonetheless, we decided to keep the upper limit of 20 questions in order to build a communication system that could be used in a reasonable time even using a ƒNIRS‑based BCI (20 seconds for each question).”